# Help seeking for intimate partner violence in a resource-constrained setting: A latent class analysis of the Nigerian demographic health survey dataset

Omolola Titilayo Alade[1*], Forough Farrokhyar[1,2,3], Sheila Ann Sprague[2,3], Anita Acai[4,5], Mohit Bhandari[1,2,3]

1 Mary Heersink School of Global Health and Social Medicine, Faculty of Health Sciences, McMaster University, Hamilton, Ontario, Canada, 2 Department of Surgery, Faculty of Health Sciences, McMaster University, Hamilton, Ontario, Canada, 3 Department of Health Research Methods, Evidence and Impact, McMaster University, Hamilton, Ontario, Canada, 4 Department of Psychiatry and Behavioural Neurosciences, McMaster University, Hamilton, Ontario, Canada, 5 McMaster Education Research, Innovation and Theory (MERIT) Centre, Hamilton, Ontario, Canada

* aladeo1@mcmaster.ca

## Abstract

Help seeking for intimate partner violence (IPV) is a complex process that involves reaching out to an external party. Women in resource constrained settings face unique constraints when seeking help for IPV but the latent classes of their help seeking behaviour in IPV has not been described. We therefore conducted a latent class analysis of help seeking behaviour among women experiencing IPV in Nigeria using the nationally representative 2018 Nigeria Demographic Health Survey (DHS) data. Nigeria was selected as an example of a resource constrained setting because close to half of its population is multidimensionally poor with significant financial and service barriers. Help seeking was defined by the latent class indicators of the places where or people from whom women sought help. The data were analysed in MPlus version 8.10 and survey sampling weights were applied. The relative fit of the models was compared using Bayesian Information Criterion (BIC), Adjusted BIC (ABIC), Lo-Mendell-Rubin Likelihood Ratio Test (LMR) $p$-values, and entropy values. Of the 3,054 women who experienced physical or sexual violence, 1,041 (33%) women reported seeking for help and a four-class model of help seeking behaviour (BIC=3910.80, ABIC=3837.70, LMR $p$-value=0.0002, and entropy value=0.92) was described: Class I (Own Family; 49%), Class II (Everywhere; 18%), Class III (Predominantly Formal; 5%), and Class IV (Predominantly Partner's Family; 28%). Women evinced a high reliance on informal sources for help. However, women with a history of sexual violence were most likely to access formal sources of help. Interventions for IPV have focussed on formal services but in resource constrained settings, the focus needs to be redirected to interventions for empowering informal sources of help (family, friends and neighbours) without neglecting formal systems.

**Data availability statement:** The data are owned by a third-party organization who do not permit the authors to share the data. Requests for the data may be sent to https://dhsprogram.com/data/available-datasets.cfm.

**Funding:** The author(s) received no specific funding for this work.

**Competing interests:** The authors have declared that no competing interests exist.

## Introduction

Intimate partner violence (IPV) is a devastating experience that disproportionately affects women. Globally, about 27% of women aged 15 years and older (641 million) have been victims of IPV at least once in their lifetime [1]. When categorized by the United Nations' economic classification, the highest prevalence of lifetime experience of intimate partner violence was recorded in the Least Developed Countries (37%), while the lowest rates were observed in high income countries (16–23%) [2]. The IPV landscape in Nigeria reflects the global picture with 17–24% of ever married/partnered women aged 15 years and older reporting at least one exposure to IPV in their lifetime [3,4].

One of the ways women experiencing IPV respond is to actively reach out for assistance. This is referred to as help seeking. Help seeking is both a cognitive and behavioural process characterized by "problem focus, intentional action and interpersonal interaction with a third party" [5–7]. It is an active process that begins with the recognition of the violence situation as a problem, the realization that self-resources are inadequate to cope with the problem and the need for intervention by an external agent [5,7,8]. Although help seeking is an important coping strategy in IPV it is underutilized in resource constrained settings with a reported average prevalence of 35% in low- and middle-income countries (LMIC) [2]. A study in Nigeria found that only about 40% of women experiencing IPV sought help to stop the perpetrator from hurting them again [9]. The same study further found that help seeking behaviour varied widely across the states in Nigeria, ranging from 12% to 65%.

The predictors of help seeking behaviour have been categorized as individual, state, and national factors [9]. Individual factors include attitudes towards IPV, empowerment of the woman, as well as the severity and type of violence. State and national factors include human development indices derived from a composite of literacy rates, per capita gross domestic product, life expectancy, and legal policies on IPV such as the definition of rape and state interference with domestic issues. Another study in Nigeria specifically highlighted the type and severity of IPV as predictors of help seeking [10]. They found that women experiencing physical violence were more likely to seek help than those experiencing sexual violence. This was attributed to the culture of silence and stigma associated with sexual acts. They further observed that severe IPV was more likely to result in help seeking.

Individual strategies (behaviour change, negotiating, and resisting) may be the first response to IPV but this may progress to help seeking from external sources including informal (family, friends and neighbours) and formal sources (shelters, police, legal, and medical services) [5,11]. It is influenced by a complex interplay of individual and contextual factors including the type and severity of the IPV and may not occur due to barriers in access (financial, cultural, geographic) [11]. In high-income countries, help for women experiencing violence includes social programs and formal services [11]. On the other hand, support systems for women experiencing IPV in resource-constrained settings are much less structured and largely rely on informal sources [12]. For example, in Nigeria, 90% of the women who sought help for IPV did so from informal sources (their own family, partner's family, neighbours and friends)

while only 7.5% of them sought help from formal institutions (healthcare institutions, legal services, social organizations, and police services) [13].

The latent classes of help seeking for IPV in resource-constrained settings remain unexplored. It is important to fill this research gap and direct targeted interventions to improve help seeking in these settings [12]. Therefore, this study aimed to conduct a latent class analysis (LCA) of help seeking behaviour among women experiencing IPV using data from the most recent Nigerian Demographic Health Survey, with Nigeria being an example of a resource-constrained setting.

## Resource constrained settings

Resource constrained or low resource settings describe the complex interplay of inequities which may occur in any country of the world. Other terms that have been used include global north/global south or developed/developing countries, but these falsely suggest that inequities are confined to certain regions of the world. In a scoping review, van Zyl et al demonstrated that 17% of articles about resource constraints in rehabilitation health were in "high income" countries. This finding suggests that resource constrained settings are not limited to any region of the world but are found wherever there is a conflation of inequitable access to financial resources, healthcare service, infrastructure, research outputs, social and/or human resources [14]. These access issues may be further compounded by geographical and environmental barriers, cultural beliefs and practices and inefficient policies.

According to the World Bank, Nigeria is classified as a lower middle-income country by its gross national income per capita [15]. It is not uniformly characterized by resource constraint (no country is) but is used in this paper as an example of a resource constrained setting because of its relatively high per capita gross domestic product of US$1,597 compared to its poverty indices [15]. Close to half of the population in Nigeria is multidimensionally poor or vulnerable to multidimensional poverty [16]. Multidimensional poverty reinforces and exacerbates the earlier identified characteristics of resource constrained settings.

## Theoretical framework

The current study was based on the theoretical framework of help seeking behaviour in IPV described by Liang et al [5]. This framework coupled with the socio-ecological model [17] outline individual, interpersonal, and socio-cultural influences on women's help seeking as they move non-linearly from recognising intimate partner violence as a problem to taking a decision to seek help and choosing a source of help. Liang's model focussed on the cognitive process involved in help seeking for IPV using a population of South Asian women in the USA. Ultimately, this focus on cognition has served to lend the theory wide application [18]. Other theories have been used to conceptualize help seeking in IPV among women, including Waller's theory of help seeking behaviour described among African American women survivors of IPV [6]. However, Waller's theory centred on racial discrimination and the critical role of the black church which may limit its application to women in Nigeria [6].

The factors selected for this study were limited by the availability of the data in the DHS dataset. The individual influences on women's help seeking for IPV were assessed by personal characteristics including age, presence or absence of children, level of education, intergenerational history of IPV, attitude towards IPV, and type and severity of IPV. The interpersonal factors are those factors which influence the relationship with an intimate partner and were assessed by the presence or absence of other wives or partners in the relationship, the woman's marital status, and family connections. Socio-cultural factors are external to the relationship but may shape help seeking for IPV and were assessed by region of residence within the country, urban or rural setting, religion, access to healthcare, and wealth index.

## Methods

This was a latent class analysis (LCA) of the 2018 Nigeria Demographic Health Survey (DHS) data, the most recently available DHS survey dataset in Nigeria. The DHS program is an international collaborative effort primarily funded by the United States Agency for International Development (USAID) to conduct nationally representative health surveys in more

than 90 low- and middle-income countries for about four decades [19]. It was conceived to provide high quality, comparable data across countries for the purpose of planning, monitoring, and evaluating population, health, and nutrition programs [19].

In Nigeria, these surveys are implemented by the National Population Commission of Nigeria and were first conducted in 1990 [13]. The survey was implemented using computer-assisted personal interviewing to collect data in person. Participants (women and men) were recruited at the household level across Nigeria and the data collection took place from 14 August to 29 December 2018. The dataset was downloaded for analysis on 16 September 2023.

## Study location

Nigeria is a country in West Africa with an estimated population of 200,000,000 people [20]. It is made up of six geo-political zones further divided into 36 states and the Federal Capital Territory, Abuja [13]. These states are further divided into Local Government Areas (LGAs) to total 774 LGAs. LGAs are further divided into enumerations areas (EAs) for a total of 662,529 EAs with an average of 856 EAs per LGA. An EA is the smallest area into the which the country is demarcated by the National Population Commission of Nigeria for census purposes [21].

In 2023, the World Bank estimated that 54% of the Nigerian population lived in urban areas [22]. Based on estimates in 2018, data from the World Bank show that only 3.3% of the population in Nigeria is covered by a social insurance program [23]. Concerning education, 8.1% of the population (aged 25 years or older) is estimated to have a Bachelor's degree or equivalent with a distribution of 10.6% among males and 5.8% among females [23]. In relation to IPV, 28% of women were estimated to accept that a man was justified in beating his wife/female partner if she did at least one of 5 things – argued with him, went out without informing him, refused to have sex with him, burnt food or neglected their children [23].

## Sampling strategy

A stratified, two-stage cluster design was used in the DHS survey. Enumeration areas (EAs) were stratified into urban and rural areas as the sampling units for the first stage. At this first stage, 1,400 EAs were randomly selected. The second stage was a complete listing of households in the 1,400 EAs, out of which 41,688 households were randomly selected by computer programming (computer generated random numbers). Of these selected households, 40,666 were occupied and people in 40,427 of these were interviewed (in person), yielding a response rate of 99%.

The participants in the households were women aged 15–49 years and men aged 15–59 years. The men's survey was conducted in one-third of the sample households and included all men aged 15–59 years present in these households. In this subsample of households where men were interviewed, one eligible woman in each household was randomly selected to be asked additional questions about domestic violence. In total, 10,678 women were surveyed about their experiences with domestic violence. Only 8,910 women who were ever married or partnered of the 10,678 women were asked questions on intimate partner violence (IPV). Of the 8,910 ever married/partnered women, 6007 (66%) reported that they had experienced at least one form of IPV (control issues, emotional, physical or sexual violence) while 3,054 women reported physical and/or sexual violence only. Only these 3,054 women who reported that they experienced physical and/or sexual IPV were asked if they sought help for IPV, of which 1,041 women reported that they had (Fig 1).

## Outcome and exposure variables

**Intimate partner violence (IPV).** IPV was defined as at least one form of control issue, emotional, physical, or sexual violence from a romantic partner in the past year [12,24]. IPV was categorized as a binary variable (0 = No and 1 = Yes) from variables in the dataset that measured control issues, emotional violence, physical violence, and sexual violence. Any woman that reported at least one control issue or emotional, physical, or sexual violence in the past year was categorized as having experienced IPV.

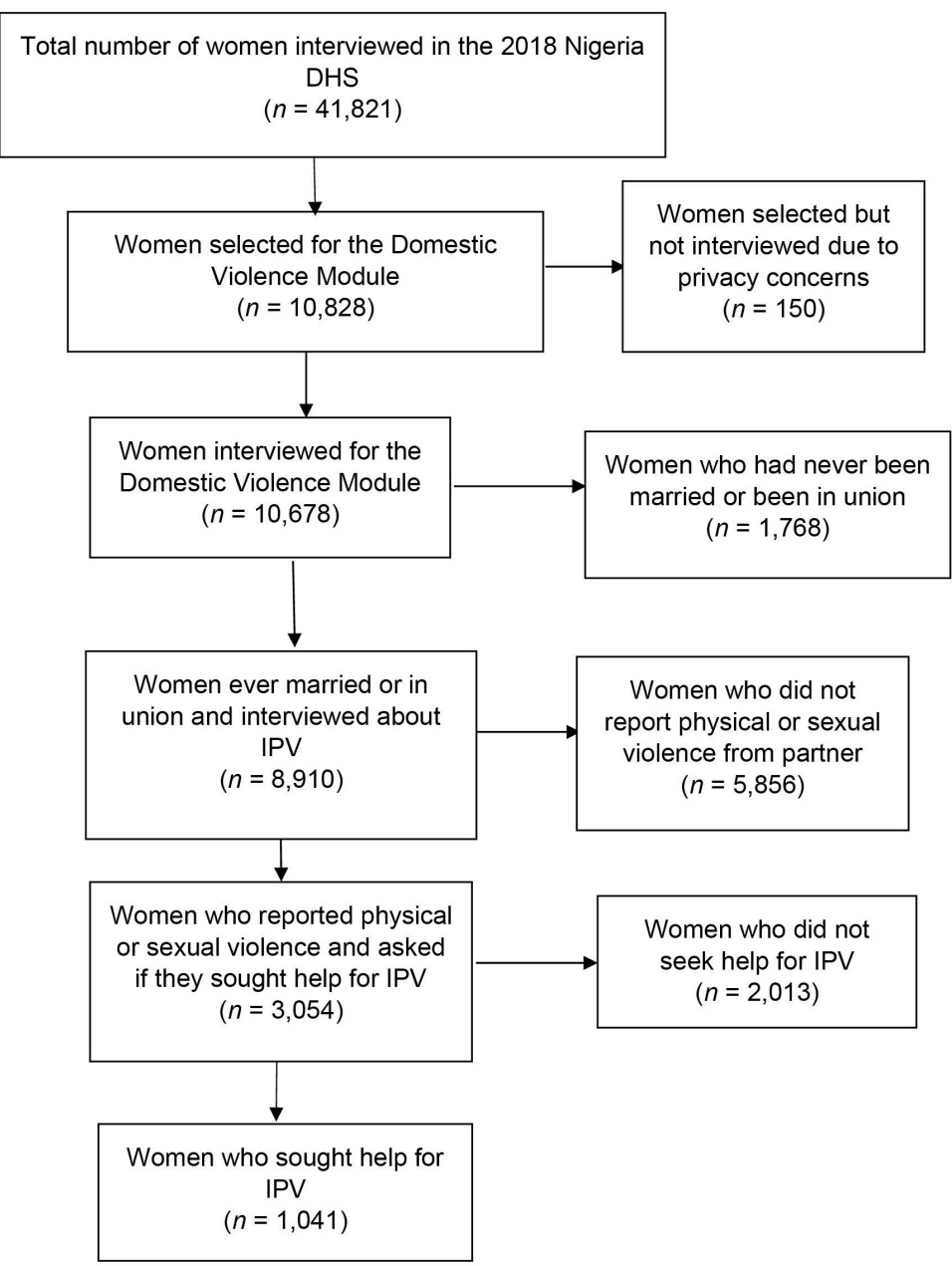

**Fig 1. Flow diagram of women included in the LCA.** IPV – Intimate partner violence.

## Help seeking for IPV

Help seeking for IPV, the outcome variable, was asked only of women who reported experiencing physical and/or sexual violence and was determined from the question, "Thinking about what you yourself have experienced among the different things we have been talking about, have you ever tried to seek help to stop the person (s) from doing this to you?" Those who answered yes were asked about the sources or people from where/whom they sought help. The options were their

own family, their husband or partner's family, a friend, a neighbour, a religious leader, doctor or other healthcare provider, the police, a lawyer, a social service organization, and other source not previously listed.

The socio-demographic variables were the ages in years of the woman and her partner, number of children the woman had, geographic region of residence in the country (North Central, North East, North West, South East, South South and South West), place of residence (urban or rural), highest educational level of the woman and her partner (categorized as none, primary, secondary, and tertiary education), religion, wealth index, marital status, and number of other wives the woman's husband or partner had, attitude towards IPV (accepting if a woman responded, "yes" or "I don't know", to at least one of the following prompts - "Beating justified if wife goes out without telling husband", "Beating justified if wife neglects the children", "Beating justified if wife argues with husband", "Beating justified if wife refuses to have sex with husband" and "Beating justified if wife burns the food") and intergenerational history of IPV. Intergenerational history of IPV was categorized as "Yes", "No" or "I don't know" from respondent's recorded response to, "Father of respondent ever beat her mother." Visit to a healthcare facility in the past 12 months was a binary variable categorized as yes or no.

The wealth index is a household level index that was calculated by principal component analysis from household characteristics such as flooring material used in the house, sanitation facilities and presence or absence of consumer goods such as refrigerators, transport vehicles in the household. Each person in the household got the same wealth score. These scores were then ranked in a distribution table and divided into quintiles. Therefore, women with a wealth index of 1 were members of a household with a wealth score in the top 20% of the wealth score distribution while women with a wealth index of 5 had household scores in the bottom 20% of the wealth score distribution [25].

A woman was categorized as having experienced less severe violence if she responded, Yes, to at least one of the following prompts: "Ever been pushed, shook or had something thrown by husband/partner", "Ever been punched with fist or hit by something harmful by husband/partner" or "Ever had arm twisted or hair pulled by husband/partner" while severe violence was recorded if a women responded, Yes, to at least one of these prompts: "Ever been kicked or dragged by husband/partner", "Ever been strangled or burnt by husband/partner" or "Ever been threatened with knife/gun or other weapon by husband/partner" [25].

The indicator variables for the LCA were the places or people from which women experiencing physical or sexual violence sought help. These were categorized as 1) their own family, 2) their partner's family, 3) their neighbours, 4) their friends, and 5) formal sources. The formal source category was created by collapsing help seeking from a religious leader, doctor or other healthcare provider, the police, a lawyer, or a social service organization into a single category because only 74 women sought help from these sources. Similar recategorization has been described in previous studies using DHS data [9,26].

## Data analysis

Sampling weights were used in the analysis to account for the complex survey design and ensure that the results were representative of the target population. The weights were provided in the DHS dataset and used to adjust for unequal probabilities of selection and non-response. These sampling weights were applied to both descriptive statistics using R statistical software, using the survey package with functions *svydesign*, *svymean*, *svyquantile*, *svyby*, *svyttest*, and *svychisq* [27]. LCA using Mplus version 8.10 [28] with "mixture complex" as the analysis type command was also performed. "Mixture complex" allowed for the incorporation of weight, primary sampling units and strata in the analysis.

The prevalence of IPV was estimated by dividing the number of women who had at least one IPV experience in the last year by the total number women who were surveyed for this question. The prevalence of seeking help for IPV was estimated by dividing the number of women who sought help for IPV by the total number of women who experienced physical and/or sexual IPV.

For the LCA, the relative fit of the models was compared using Bayesian Information Criterion (BIC), Adjusted BIC (ABIC), Lo-Mendell-Rubin Likelihood Ratio Test (LMR) *p*-values, and entropy values. Multiple models with two to five class

solutions were created. Lower BIC or ABIC values indicated a better model while the LMR *p*-values indicated if a model fit better than a model with one less class. A *p*-value of less than .05 suggested that the model with a higher number of classes fit better. Entropy values range from zero to one, with values greater than 0.60 indicating a good model fit [29]. Lastly, the models were assessed on clear visual demarcations and meaningfulness of the classes [30].

Within the selected best class model, multinomial logistic regression was conducted in a post-hoc exploratory analysis to determine the factors associated with class membership using the R3Step model in Mplus version 8.10 [31]. The selected variables for this analysis were based on findings in previous studies: age of the woman, region, residence, highest level of education, wealth index, previous visit to a health facility, severity of violence, type of violence, intergenerational history of IPV, and the woman's attitudes towards IPV [5,10,32].

### Ethical considerations

Permission to use the publicly available dataset in this analysis was requested and granted by the demographic and health survey (DHS) program (https://dhsprogram.com/Data/). The DHS Program and the National Health Research Ethics Committee of Nigeria reviewed and approved the research protocol [25]. Each respondent was read an informed consent statement and could verbally accept or decline to participate with no consequences. Parents or guardians provided consent for adolescents. The consent decision was witnessed and documented on the data collection forms by the interviewers. The data were only made publicly available after they were fully anonymized. The Hamilton integrated research ethics board of McMaster University further gave ethical approval for the study (Project ID: 17782).

### Results

A total of 41,821 women aged 15–49 years were interviewed as part of the DHS. Of these, 10,678 women were interviewed about their experiences of domestic violence. Questions about intimate partner violence (IPV) were asked of 8,910 women who were ever married or partnered (Fig 1).

These 8,910 women had a mean age of $32.2 \pm 11.33$ years. The women had 0–12 children with 7% having no children and 66% having one to four children. The median number of children was three with an interquartile range of two to five. According to the survey data, 66% (6007/8,910) of women who had ever been married or partnered reported that they had experienced at least one form of IPV.

Table 1 (complete version S1 Table in the appendix) shows that the women who experienced IPV were similar in age and number of children to those who did not. Among women who experienced IPV, the women from the North West region of the country had the highest proportion (30.8%) while those the South East had the least proportion (11.3%). Women living in rural areas were more likely to experience IPV (60.4%) compared with those living in urban areas (39.6%). About 40% of the women who experienced IPV had no formal education. More than half of the women who experienced IPV were of the Islamic religion (55.5%). The proportion of women experiencing IPV appeared to be evenly distributed across the quintiles of the wealth index and visit to a health facility in the last 12 months.

Slightly more than one third (34%) of the 3,054 women who experienced physical or sexual IPV reported that they sought help. Fig 2 shows the distribution of the people/places from whom/which they sought help. More women sought help from informal sources compared with formal sources.

Table 2 shows that the characteristics of women who sought help for IPV were similar to those who did not. However, the factors that were significantly associated with women seeking help for IPV were region of the country, marital status, woman's attitude towards IPV and intergenerational history of IPV. More women sought help for IPV from the South East and South South regions compared to other regions, married women were more likely to seek help than women in other marital situations, women who were non-accepting of IPV were more likely to seek help than those who were accepting of IPV and women with no intergenerational history of IPV were more likely to seek help than those with an intergenerational history of IPV.

**Table 1. Background characteristics of women who experienced IPV compared with those who did not (*n* = 8,910).**

| Characteristic | Total Study Population (*n* = 8,910) | Women who experienced IPV (*n* = 6007) | Women who did not experience IPV (*n* = 2903) | *p*-value |
|---|---|---|---|---|
| Age (Years)& | 32.2 (11.33) | 31.9 (13.53) | 32.8 (19.14) | $= 4.7 \times 10^{-4}$ |
| Number of Children& | 3.5 (3.78) | 3.5 (4.00) | 3.4 (6.50) | $= 2.5 \times 10^{-1}$ |
| Region# | | | | |
| North Central | 14.0 | 15.7 | 10.8 | $<2.2e^{-16}$ |
| North East | 15.4 | 18.7 | 9.1 | |
| North West | 29.7 | 30.8 | 27.6 | |
| South East | 11.5 | 11.3 | 12.0 | |
| South South | 10.8 | 12.1 | 8.4 | |
| South West | 18.5 | 11.4 | 32.1 | |
| Place of Residence# | | | | |
| Urban | 43.3 | 39.6 | 50.7 | $= 3.995e^{-11}$ |
| Rural | 56.7 | 60.4 | 49.3 | |
| Highest Educational Level# | | | | |
| No Education | 41.1 | 43.0 | 37.6 | $= 1.503e^{-08}$ |
| Primary | 16.6 | 17.2 | 15.3 | |
| Secondary | 32.8 | 32.0 | 34.3 | |
| Higher | 9.5 | 7.8 | 12.8 | |

&Mean and Standard Deviation.

#Percentages (these may not add up to 100 due to rounding).

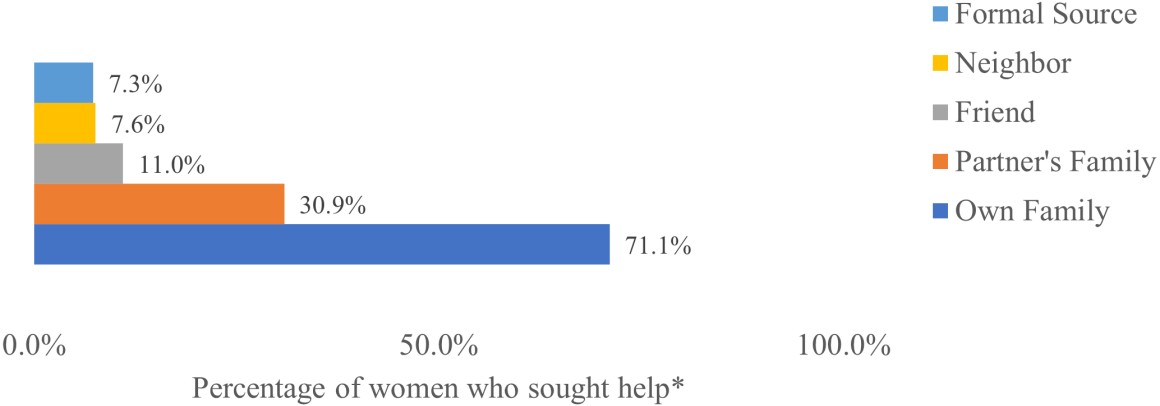

■ Formal Source
■ Neighbor
■ Friend
■ Partner's Family
■ Own Family

Percentage of women who sought help*

**Fig 2. Sources where women experiencing IPV sought help (*n* = 1,041).** *Percentages may exceed 100 as respondents were allowed to select multiple answers.

Table 3 shows the latent class models that were run to categorize help seeking behaviour among Nigerian women who experienced physical or sexual violence and sought help for it. The five-class latent model had the lowest BIC and ABIC values (3905.90 and 3813.79), respectively. However, its entropy value (0.93) was similar to the four-class latent model (0.92) and not statistically different from it (*p* = 0.38). Lastly, the addition of one more class to the four-class latent model did not improve the interpretability of latent classes of help seeking behaviour among these women. Therefore,

**Table 2. A comparison of the characteristics of women who sought help for IPV compared with those who did (n=3,054).**

| Characteristic | Women who sought help for IPV (*n*=1041) | Women who did not seek help for IPV (*n*=2,013) | *p*-value |
|---|---|---|---|
| **Age (Years)** & | 32.4 (10.78) | 32.3 (10.92) | = 0.87 |
| **Number of Children**& | 3.3 (3.07) | 3.4 (3.27) | = 0.79 |
| **Region**# | | | |
| North Central | 12.2 | 22.8 | <2.2e-16 |
| North East | 20.6 | 20.6 | |
| North West | 10.1 | 12.7 | |
| South East | 21.5 | 11.6 | |
| South South | 24.5 | 12.8 | |
| South West | 11.2 | 19.4 | |
| **Place of Residence**# | | | = 0.16 |
| Urban | 47.8 | 44.1 | |
| Rural | 52.2 | 55.9 | |
| **Level of Education**# | | | = 0.12 |
| No Education | 28.2 | 34.1 | |
| Primary School | 20.2 | 18.2 | |
| Secondary School | 41.6 | 37.7 | |
| Higher | 9.9 | 9.9 | |
| **Religion**# | | | |
| Catholic | 14.6 | 11.8 | |
| Other Christian | 50.7 | 44.1 | |
| Islam | 34.2 | 43.5 | |
| Traditionalist | 0.4 | 0.5 | |
| Other | 0.0 | 0.0 | |
| **Wealth Index**# | | | = 0.07 |
| Poorest | 12.8 | 15.5 | |
| Poorer | 17.5 | 18.2 | |
| Middle | 23.0 | 21.3 | |
| Richer | 26.0 | 20.7 | |
| Richest | 20.8 | 24.2 | |
| **Visit to Health Facility in Last 12 Months**# | | | |
| Yes | 53.2 | 49.5 | = 0.14 |
| No | 46.8 | 50.5 | |
| **Marital Status**# | | | = 8.128e-06 |
| Married | 78.8 | 88.1 | |
| Living with Partner | 6.4 | 4.5 | |
| Widowed | 6.4 | 2.8 | |
| Divorced | 4.0 | 1.8 | |
| Separated | 4.4 | 2.8 | |
| **Attitude Towards IPV**# | | | = 0.03 |
| Non-Accepting | 72.8 | 67.8 | |
| Accepting | 27.2 | 32.2 | |

*(Continued)*

**Table 2.** (Continued)

| Characteristic | Women who sought help for IPV (*n*=1041) | Women who did not seek help for IPV (*n*=2,013) | *p*-value |
|---|---|---|---|
| **Intergenerational IPV History#** | | | = 0.0009 |
| No | 68.7 | 76.0 | |
| Yes | 22.7 | 16.1 | |
| Don't Know | 8.6 | 7.9 | |

&Mean and Standard Deviation.

#Percentages (these may not add up to 100 due to rounding).

the four-class latent model was selected over the five-class model, to describe the help seeking behaviour in this population, because of its parsimony, interpretability and non-statistically significant difference from (p=0.38) from the five-class model.

The four-class model is shown in Fig 3: Class I – Seeks Help from Own Family Alone (49.1%), Class II – Seeks Help from Everywhere (18.1%), Class III – Predominantly Seeks Help from Formal Sources (5.3%) and Class IV – Predominantly Seeks Help from Partner's Family (27.5%).

Fig 3 further shows the probability distribution of seeking help for IPV for women in each of the latent classes. The classes were named based on this probability distribution. The women belonging to latent Class I were characterised by having a 100% probability of seeking help from their own family and a 0% probability of seeking help from all other sources; therefore, this class was named the Own Family Alone help seeking class. The women in latent Class II were named the Everywhere help seeking class because they had a probability greater than 0 of seeking help from each of the indicator sources: 50% probability of seeking help from their own family, 25% probability of seeking help from their partner's family, 42% from neighbour, 64% from friend, and 7% probability of seeking help from formal sources. The women in latent Class III had a 100% probability of seeking help from a formal source, only an 8% probability of seeking help from their own family, and a 0% probability of seeking help from all other sources. As such, they were named the Predominantly Formal help seeking class. Lastly, women belonging to latent Class IV were named the Predominantly Partner Family help seeking class because their probability distribution of help seeking from the indicator classes were 100% from their partner's family, 45% from their own family, 2% from formal sources, and 0% from neighbours and friends.

Table 4 shows the factors associated with belonging to Class II, III and IV compared with Class I. It further shows the factors associated with membership within each class. Women in Class I (Own Family) were more likely to be from the North East of Nigeria than women in all the other classes: Class II (Everywhere), Class III (Predominantly Formal) and

**Table 3. Goodness-of-fit indices comparing class membership models of seeking help for IPV.**

| LCA Model | BIC | ABIC | LMR Comparisons of classes | *p*-value for LMR | Entropy |
|---|---|---|---|---|---|
| 2-Class | 4173.12 | 4138.18 | 1-Class vs 2-Class | 0.0000 | 0.67 |
| 3-Class | 4013.52 | 3959.53 | 2-Class vs 3-Class | 0.0000 | 0.85 |
| 4-Class | 3910.79 | 3837.74 | 3-Class vs 4-Class | 0.0002 | 0.92 |
| 5-Class | 3905.90 | 3813.79 | 4-Class vs 5-Class | 0.38 | 0.93 |

BIC – Bayesian Information Criterion.

ABIC – Adjusted Bayesian Information Criterion.

LMR – Lo-Mendell-Rubin Likelihood Ratio Test.

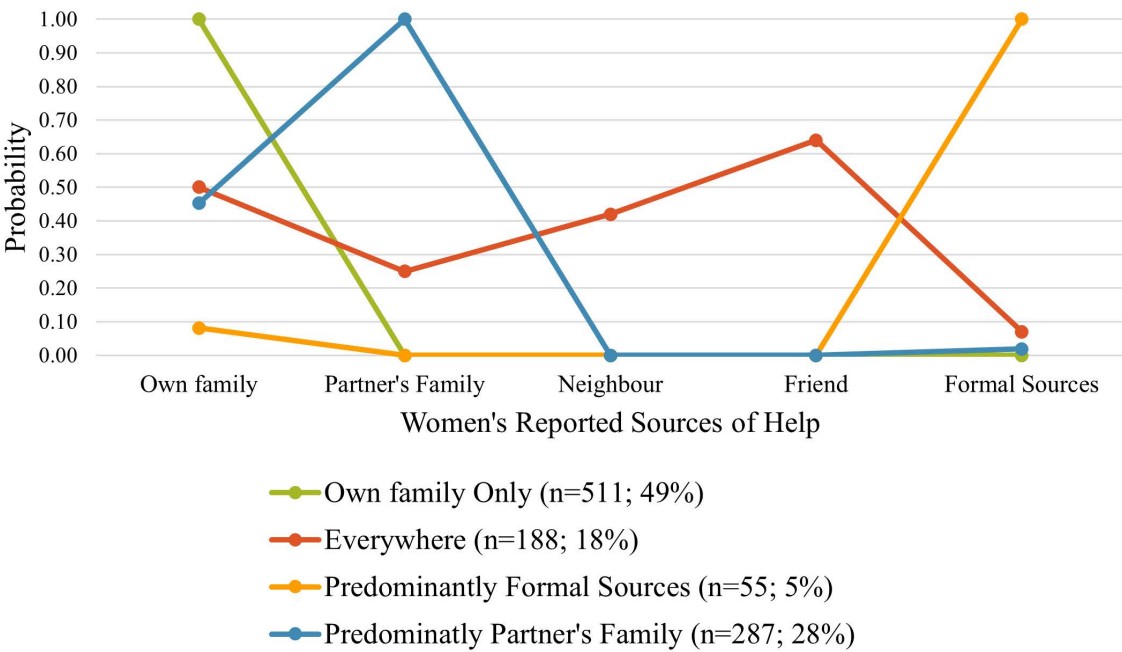

**Fig 3. Latent classes of help seeking behaviour among women experiencing IPV in Nigeria (*n* = 1,041)\*.** \*Class membership defined by a woman's probability of seeking help from each of the indicator sources.

Class IV (Predominantly Partner Family). The other differences were with the women who sought help from Predominantly Formal sources (Class III); they were less likely to be from the South South of the country but more likely to have reported a previous history of sexual violence than those in Class I. Women who went everywhere for help (Class II) were more likely to have primary education as their highest level of education compared with women in Class I (Own Family).

## Discussion

In this study, we described an LCA of help seeking behaviour among women experiencing IPV in a resource-constrained setting. We used the most recent DHS data from Nigeria as an example of a resource-constrained setting where women rely predominantly on informal sources of help for IPV. The indicators used for our analysis were 1) their own family, 2) their partner's family, 3) their neighbours, 4) their friends, and 5) formal sources and the selected model for the latent classes of help seeking was a four-class model: Own Family – 49%), Everywhere – 18%, Predominantly Formal – 5%, and Predominantly Partner Family – 28%. This is the first time that latent classes of help seeking for IPV have been delineated among women in a low-and-middle-income country context. More than 70% (77%) of the women in this survey sought help from their own or partner's family. This finding underscores the importance of directing help seeking for IPV interventions at the extended family level in these settings as previously described in the cultural identity domain of the PEN-3 model [33]. Previous community level interventions for IPV have focussed on men, women, couples or adults in a community without deliberate attempts to ensure the inclusion of the extended family members of women experiencing IPV [12,34–36].

Concerning help seeking from a formal source, only 5% of the women (Class III) predominantly sought help from a formal source and this was statistically associated with sexual violence. The relatively small proportion (5%) of women in this class agrees with the findings of a study in Ghana, which observed that women experiencing IPV in resource-constrained settings distrusted formal institutions as sources of help [37]. However, this class was associated with a history of sexual violence even

**Table 4. Factors associated with latent class membership\* (*n*=1,041).**

| | Odds Ratio | Standard Error | 95% Confidence Interval for Odds Ratio | |
|---|---|---|---|---|
| **Class II: Everywhere** | | | | |
| Age of Woman in Years | 1.01 | 0.019 | 0.97 | 1.05 |
| **Region (Reference is South West Region)** | | | | |
| North Central | 2.22 | 1.413 | 0.64 | 7.73 |
| North East | 0.15 | 0.085 | **0.05** | **0.46** |
| North West | 1.81 | 1.455 | 0.38 | 8.74 |
| South East | 0.83 | 0.378 | 0.34 | 2.03 |
| South South | 0.42 | 0.212 | 0.16 | 1.13 |
| **Residence (Reference is Urban)** | | | | |
| Rural | 1.17 | 0.362 | 0.64 | 2.14 |
| **Highest Level of Education of Woman (Reference is Higher Education)** | | | | |
| No Education | 1.75 | 0.921 | 0.62 | 4.91 |
| Primary Education | 2.76 | 1.339 | **1.07** | **7.14** |
| Secondary Education | 1.59 | 0.655 | 0.71 | 3.57 |
| **Wealth Index (Reference is Richest)** | | | | |
| Poorest/Poorer | 1.00 | 0.000 | 1.00 | 1.00 |
| Richer/Richest | 1.00 | 0.000 | 1.00 | 1.00 |
| Number of Children Woman Has | 0.87 | 0.067 | 0.75 | 1.01 |
| **Visit to a Health Facility in the Last 12 Months (Reference is Yes)** | | | | |
| No Visit | 1.04 | 0.285 | 0.61 | 1.78 |
| **Experience of Less Severe Violence (Reference is No)** | | | | |
| Yes | 1.91 | 0.638 | 0.99 | 3.67 |
| **Experience of Severe Violence (Reference is No)** | | | | |
| Yes | 1.68 | 0.506 | 0.93 | 3.03 |
| **Experience of Sexual Violence (Reference is No)** | | | | |
| Yes | 1.84 | 0.704 | 0.86 | 3.89 |
| **Intergenerational History of IPV (Reference is None)** | | | | |
| Yes/Don't Know | 1.00 | 0.00 | 1.00 | 1.00 |
| **Woman's Attitude to IPV (Reference is Non-Accepting Attitude)** | | | | |
| Accepting Attitude | 1.22 | 0.383 | 0.66 | 2.25 |
| **Class III: Predominantly Formal** | | | | |
| Age of Woman in Years | 1.01 | 0.033 | 0.94 | 1.07 |
| **Region (reference is South West Region)** | | | | |
| North Central | 0.58 | 0.609 | 0.07 | 4.54 |
| North East | 0.22 | 0.168 | **0.05** | **0.99** |
| North West | 3.53 | 3.453 | 0.52 | 23.99 |
| South East | 0.26 | 0.220 | 0.05 | 1.37 |
| South South | 0.25 | 0.170 | **0.06** | **0.96** |
| **Residence (Reference is Urban)** | | | | |
| Rural Residence | 1.36 | 0.660 | 0.52 | 3.52 |
| **Highest Level of Education of Woman (Reference is Higher Education)** | | | | |
| No Education | 0.46 | 0.375 | 0.09 | 2.29 |
| Primary Education | 0.81 | 0.718 | 0.14 | 4.62 |
| Secondary Education | 0.97 | 0.816 | 0.19 | 5.03 |

*(Continued)*

**Table 4.** (Continued)

| | Odds Ratio | Standard Error | 95% Confidence Interval for Odds Ratio | |
|---|---|---|---|---|
| **Wealth Index (Reference is Richest)** | | | | |
| Poorest/Poorer | 1.00 | 0.000 | 1.00 | 1.00 |
| Richer/Richest | 1.00 | 0.000 | 1.00 | 1.00 |
| Number of Children Woman Has | 0.91 | 0.102 | 0.73 | 1.13 |
| **Visit to a Health Facility in the Last 12 months (Reference is Yes)** | | | | |
| No Visit | 0.63 | 0.304 | 0.24 | 1.62 |
| **Experience of Less Severe Violence (Reference is No)** | | | | |
| Yes | 0.91 | 0.360 | 0.42 | 1.98 |
| **Experience of Severe Violence (Reference is No)** | | | | |
| Yes | 2.53 | 1.072 | 1.10 | 5.80 |
| **Experience of Sexual Violence (Reference is No)** | | | | |
| Yes | 3.12 | 1.522 | **1.20** | **8.12** |
| **Intergenerational History of IPV (Reference is None)** | | | | |
| Yes/Don't Know | 1.00 | 0.000 | 1.00 | 1.00 |
| **Woman's Attitude to IPV (Reference is Non-Accepting Attitude)** | | | | |
| Accepting Attitude | 1.17 | 0.573 | 0.45 | 3.06 |
| **Class IV: Predominantly Partner Family** | | | | |
| Age of Woman in Years | 0.98 | 0.017 | 0.95 | 1.02 |
| **Region (Reference is South West Region)** | | | | |
| North Central | 2.71 | 1.686 | 0.80 | 9.17 |
| North East | 0.31 | 0.182 | **0.01** | **0.98** |
| North West | 3.13 | 2.427 | 0.68 | 14.31 |
| South East | 0.46 | 0.241 | 0.16 | 1.29 |
| South South | 1.06 | 0.537 | 0.39 | 2.86 |
| **Residence (Reference is Urban)** | | | | |
| Rural Residence | 1.31 | 0.367 | 0.76 | 2.27 |
| **Highest Level of Education of Woman (Reference is Higher Education)** | | | | |
| No Education | 1.21 | 0.658 | 0.42 | 3.52 |
| Primary Education | 1.91 | 1.043 | 0.66 | 5.57 |
| Secondary Education | 1.25 | 0.596 | 0.49 | 3.18 |
| **Wealth Index (Reference is Richest)** | | | | |
| Poorest/Poorer | 1.00 | 0.000 | 1.00 | 1.00 |
| Richer/Richest | 1.00 | 0.000 | 1.00 | 1.00 |
| Number of Children Woman Has | 0.94 | 0.068 | 0.82 | 1.09 |
| **Visit to a Health Facility in the Last 12 Months (Reference is Yes)** | | | | |
| No Visit | 1.03 | 0.255 | 0.63 | 1.67 |
| **Experience of Less Severe Violence (Reference is No)** | | | | |
| Yes | 0.93 | 0.236 | 0.56 | 1.53 |
| **Experience of Severe Violence (Reference is No)** | | | | |
| Yes | 1.18 | 0.301 | 0.71 | 1.94 |
| **Experience of Sexual Violence (Reference is No)** | | | | |
| Yes | 1.88 | 0.615 | 0.97 | 3.57 |
| **Intergenerational History of IPV (Reference is None)** | | | | |
| Yes/Don't Know | 1.00 | 0.000 | 1.00 | 1.00 |

*(Continued)*

**Table 4.** (Continued)

| | Odds Ratio | Standard Error | 95% Confidence Interval for Odds Ratio | |
|---|---|---|---|---|
| Woman's Attitude to IPV (Reference is Non-Accepting Attitude) | | | | |
| Accepting Attitude | 1.26 | 0.339 | 0.75 | 2.14 |

*Reference Class is Class I (Own Family Only).

S2 Table showing the conditional probabilities and assigned latent classes for the women is in the appendix.

though women experiencing sexual violence are the least likely to seek help [10]. This finding suggests that while women who experience sexual violence are less likely to seek help than those experiencing other forms of violence, those that do seek help are more likely to seek help from formal as opposed to informal sources. These findings suggest socio-cultural links between family ties or connections, sexual violence, and help seeking from formal sources that would be worthy of further exploration.

The latent classes of help seeking behaviour in IPV, in extant literature, have been either two-class [38] or three-class models [32,39,40]. The four-class model described in this study is attributable to a delineation of family as a source of help emerging as two distinct classes—the women's own family and their partner's family.

It was difficult to make comparisons with other studies in which latent classes of help seeking for IPV were described because the indicators used for the latent classes are not standardized across studies [32,38–40]. In a study conducted in Canada, an analysis of the latent classes of help seeking for IPV using national survey data employed the following indicators – (1) Police (2) Social service agency (3) Psychiatrist/doctor (4) Family/friend. The latent classes described were – Minimal Help Seeking (39%), Seeking help from family and Friends (41%) and Substantial Help Seeking (20%) [40].

In another survey that included the use of formal and informal sources, Hu et al. conducted a non-probability sampling of heterosexual women in China, who had experienced IPV in the past year [38]. The indicators were (1) friends, (2) family members, (3) teachers, (4) neighbors, (5) the police, (6) the Women's Federation (WF) (7) mental health professionals, (8) lawyers, (9) non-profit service organizations, (10) mental health hotline, (11) medical support, (12) social media/journalists, and (13) courts and 2 classes were described – Friends/family only (86.1%) and Extensive sources (13.9%).

Other LCA studies on help seeking for IPV among women were conducted among women who used formal services in Israel and the US [32,39]. The study in Israel had 15 indicators – Use of (1) hospital emergency rooms, (2) centers for violence prevention and treatment (community-based centers for the treatment of violence), (3) women's shelters, (4) emergency hotlines, (5) social services bureaus, (6) child protection social workers, (7) centers for single mothers (centers that provide information regarding what the rights of single mothers are and also provide single mothers with individual and group counseling), (8) mental health centers (treatment and rehabilitation centers under the auspices of the health ministry), (9) legal assistance, (10) the courts, (11) police departments, (12) mother-and-child health clinics, (13) school counselors, (14) psychological services for children provided by the education ministry, and (15) women's organizations and the classes were – Minimal Use of all services (35%), Substantial Use of all services (9%) and Welfare and Criminal System Use (56%) [32]. The study in the US employed 7 indicators – (1) engaging informal networks, (2) seeking help for their abusive partner, (3) staying in a shelter, (4) seeking legal services, (5) engaging police, (6) seeking housing assistance to move to a safe location, and (7) visiting a health care provider and the classes were – Primarily engages informal networks (15%), Broadly engages formal and informal networks (50%) and Broadly engages networks but avoids legal systems (35%) [39].

Secondary findings showed that 66% of women who were ever married or had a partner had reported at least one form of emotional, physical, or sexual violence from their partner in the 12 months prior to the survey. Of these women, only 33% of those who reported physical and/or sexual violence from their partner reported that they sought help for the violence.

The covariates significantly associated with these latent classes were region of the country, previous sexual violence, and the woman's highest level of education.

Contrary to other studies [10,11,37] severity of violence was not significantly associated with help seeking behaviour. This may be explained by the definition of severity of violence in this study as one or a combination of the actions of kicking, dragging, strangling, burning or threatening with a knife, gun, or other weapon. This reductive definition fails to acknowledge the individual, interpersonal, and socio-cultural influences on women's definition of severe IPV and their decision to seek help for it [5]. In contrast, Tenkorang et al. used a principal component analysis technique to produce a weighted summative index, which may have been a more sensitive measure for severe violence [10]. Similarly, in a qualitative study among women experiencing IPV in Ghana, severe violence was defined as a perception of being in danger of death [37]. This perception would be inadequately captured by the binary categorization of the listed actions in this study.

### Study limitations

We relied exclusively on secondary data from the DHS, which may not be detailed enough to capture the nuances of help seeking behaviour in a resource-constrained setting. The complexity of the process of help seeking in IPV was reduced to latent classes of behaviour constructed using binary variables developed from a pre-determined list of sources of help. There may be places, people, and resources beyond this list that can only be explored and delineated qualitatively. Nevertheless, the latent classes described in this study are important as a foundation for future studies.

### Implication of the study

The implications of the study findings are the need for multi-faceted interventions that educate and empower both lay members of the community and workers in formal institutions to identify and respond appropriately to IPV. Interventions have often focused on formal settings such as hospitals, healthcare workers, and justice system workers but the heavy reliance on family and friends as support for women experiencing IPV suggests that community approaches to IPV prevention and response should also address these informal sources of help [12].

### Conclusion

This study has delineated the latent classes of help seeking among women experiencing IPV in a resource-constrained setting. The women evinced a high reliance on their own and partner's families for help. Women with a history of sexual violence were most likely to access formal sources of help but least likely to seek help from family. Both informal and formal systems need to be strengthened to ensure that women experiencing IPV receive the help that they need. Formal systems strengthening should specifically include sexual IPV prevention and response without neglecting other forms of IPV.

### Supporting information

**S1 Table. Background characteristics of women who experienced IPV compared with those who did not (n = 8,910).** Complete version of Table 1.
(DOCX)

**S2 Table. Conditional Probabilities and Assigned Latent Classes (n = 1,041).**
(DOCX)

### Author contributions

**Conceptualization:** Omolola Titilayo Alade.

**Data curation:** Omolola Titilayo Alade.

**Formal analysis:** Omolola Titilayo Alade.

**Methodology:** Omolola Titilayo Alade, Forough Farrokhyar, Sheila Ann Sprague, Anita Acai, Mohit Bhandari.

**Software:** Omolola Titilayo Alade.

**Supervision:** Forough Farrokhyar, Sheila Ann Sprague, Anita Acai, Mohit Bhandari.

**Writing – original draft:** Omolola Titilayo Alade.

**Writing – review & editing:** Omolola Titilayo Alade, Forough Farrokhyar, Sheila Ann Sprague, Anita Acai, Mohit Bhandari.

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
