## [Decision Letter · Decision Letter 0]

9 Apr 2025

Dear Dr. Alade,

Thank you for submitting your manuscript to PLOS ONE. After careful consideration, we feel that it has merit but does not fully meet PLOS ONE’s publication criteria as it currently stands. Therefore, we invite you to submit a revised version of the manuscript that addresses the points raised during the review process.

We look forward to receiving your revised manuscript.

Kind regards,

Obasanjo Bolarinwa, PhD

Academic Editor

PLOS ONE

Journal Requirements:

Reviewers' comments:

Reviewer's Responses to Questions

**Comments to the Author**

1. Is the manuscript technically sound, and do the data support the conclusions?

Reviewer #1: No

Reviewer #2: Yes

Reviewer #3: Yes

2. Has the statistical analysis been performed appropriately and rigorously?

Reviewer #1: Yes

Reviewer #2: Yes

Reviewer #3: Yes

3. Have the authors made all data underlying the findings in their manuscript fully available?

Reviewer #1: Yes

Reviewer #2: Yes

Reviewer #3: Yes

4. Is the manuscript presented in an intelligible fashion and written in standard English?

Reviewer #1: Yes

Reviewer #2: Yes

Reviewer #3: Yes

Reviewer #1: Authors conducted a LCA of IPV help-seeking (HS) in Nigeria. The paper has great merit, particularly in areas where little is known about available resources and support. While this paper is of great importance, it could be further strengthened by providing readers of a conceptualization of the terms, a synthesis if extant literature, identifying gaps in knowledge, better organizing the justification for this scholarship. It’s my understanding that the terms “domestic violence” is used in Nigeria rather than “intimate partner violence.” Since this is the case, I am curious how come authors did not use the terminology that is used in that country? Since authors fail to provide enough background information about IPV and underpinning dynamics for readers to contextualize the findings. Authors may also refer to the publication for organization as it is currently amiss of what is typical for academic journals. Importantly, authors discuss the 4-class model; however, the best model is actually the 5-class model, based on the BIC and entropy.

Reviewer #2: The study provides relevant and timely insights into the underlying drivers of IPV. The authors provided an extensive review of existing evidence and statistically modelled the level of class membership with multiple models which further enriched the research. The goal of these types of studies is to translate research to practice by using evidence to inform policy, strategies and initiative to reduce IPV risks and improve intervention outcomes. It would be great for the authors to consider aligning the LCA with existing models such as the Socioecological model which attempts to align critical determinants at different levels of engagement. Other alrnative models could be considered.

Furthermore, the use of "he;p' was applies loosely and insome cases interchangeably with 'reporting IPV'. There is need to unpack the meaning of 'help' - is it only talking about IPV to someone or does it go beyond this to include counselling counselling, protective & legal support, social protection, healthcare services, etc.

There is need to also explore the reason why IPV help is not sought (perhaps this is beyond the scope of this literature). But will be helpful in closing the loop.

Reviewer #3: The scientific rationale for the study was explained explicitly. The methodological gap and objective of the study was clearly stated. The design, setting, variable for the study, size and sampling techniques were outlined. Kindly state any efforts to mitigate bias.

The result and discussion were summarised. Kindly state the limitation of the study in a sub-heading of it own. State the conclusion as a sub-heading of it own and ensure it is comprehensive. State the implication of the study as a sub-heading

Summary of recommendation

1. State efforts to mitigate bias

2. State the limitation of the study

3. State the policy implications of the result findings

4. State the conclusion of the study

**Do you want your identity to be public for this peer review?** For information about this choice, including consent withdrawal, please see our Privacy Policy

Reviewer #1: No

Reviewer #2: **Yes: ** Imuwahen Victor Igharo

Reviewer #3: **Yes: ** Chukwudeh Stephen Okechukwu

---

## [Author Response · Author response to Decision Letter 1]

30 Sep 2025

Please find below a point-by-point response to each comment from the academic editor.

Updated Financial Disclosure: Not applicable.

Academic Editor’s Comments:

Response: This has been done.

2. We note that you have indicated that there are restrictions to data sharing for this study. PLOS only allows data to be available upon request if there are legal or ethical restrictions on sharing data publicly.

Response: The data are owned by a third-party organization who do not permit the authors to share the data. Requests for the data may be sent to https://dhsprogram.com/data/available-datasets.cfm.

---

## [Editor Report · Decision Letter 1]

6 Oct 2025

Help seeking for intimate partner violence in a resource-constrained setting: A latent class analysis of the Nigerian demographic health survey dataset

PONE-D-24-53009R1

Dear Dr. Alade,

We’re pleased to inform you that your manuscript has been judged scientifically suitable for publication and will be formally accepted for publication once it meets all outstanding technical requirements.

Kind regards,

Obasanjo Bolarinwa, PhD

Academic Editor

PLOS ONE
---

## [Editor Report · Acceptance letter]

PONE-D-24-53009R1

PLOS ONE

Dear Dr. Alade,

I'm pleased to inform you that your manuscript has been deemed suitable for publication in PLOS ONE. Congratulations! Your manuscript is now being handed over to our production team.

Kind regards,

on behalf of

Dr Obasanjo Bolarinwa

Academic Editor

PLOS ONE